# Structural insights into C3 convertase activity of the classical pathway of complement

Karla I. De la O Becerra [1], T. Harma C. Brondijk [1], Itziar Serna Martin[1,2] & Piet Gros [1] ✉

Immune protection by the complement system depends on C3 cleavage by C3 convertases that is critical to all three activation pathways. Structural data on convertase formation in the classical pathway and on C3-substrate binding to convertases is lacking. We present the cryo-EM structures of the pro-convertase (C4b2), convertase (C4b2b), and convertase-substrate complex (C4b2b-C3) of the classical pathway. The data show that C2 and C4b form proconvertases and convertases like factor B and C3b of the alternative pathway. Substrate C3 binds C4b of the convertase through two interfaces: one also found in the SCIN-inhibited C3bBb dimer, and another facilitated by conformational changes in C3. Bending of C3 and swinging of the C2 protease bring the C3-scissile loop into the active site. The second, charged, C4b-interaction site favors C3- substrate binding, but upon cleavage repels product C3b. Thus, a charge switch-over mechanism effects the catalytic turnover of the convertases producing opsonin C3b.

The humoral complement system in humans has important roles in tissue homeostasis and immune protection against infectious agents[1,2]. A central and pivotal step in the complement response is labelling targeted cells and debris for clearance by deposition of C3b through C3 convertases that cleave C3 into anaphylatoxin C3a and opsonin C3b[2,3]. C3 convertases are irreversibly dissociating, bimolecular proteolytic complexes with ca. 1–1.5 min. half-life times[4,5]. They are generated on surfaces by danger-recognition complexes of the (so-called) classical and lectin pathways, and through a feed-back amplification loop of the alternative pathway[1,6,7]. The initiation complexes of the classical and lectin pathways first cleave complement C4 into anaphylatoxin C4a and C4b, which binds covalently to the targeted surface through its thioester moiety[8,9]. Next, pro-enzyme C2 binds to C4b forming a C3 proconvertase, C4b2. Subsequently, C2 is cleaved by the initiation complexes yielding a C3 convertase, C4b2b[4]. In the alternative pathway, C3 proconvertases are formed by binding factor B either to fluid-phase C3(H2O), as part of the so-called 'tick-over' process, or to surface-bound C3b, as the first step in the feed-back amplification loop. Subsequently, these proconvertases are cleaved by

soluble factor D yielding C3 convertases, $C3(H_2O)Bb$ and $C3bBb$[5,10,11]. Thus, short-lived C3 convertases are formed on surfaces, providing a short, localized boost in production of C3b molecules that opsonize the targeted surface for immune clearance. Host cells are protected from opsonization by regulators that bind C3b or C4b to disrupt the bimolecular C3 convertases and degrade C3b and C4b[7,12–14]. Dysregulation of this protection causes a range of pathologies occurring in a variety of diseases[15–22], which underscores the critical role of opsonization by the C3 convertases.

C4 and C2 of the classical and lectin pathways and C3 and factor B of the alternative pathway are homologous protein pairs with amino-acid sequence identities of 30% for human C3 and C4, and 39% for factor B and C2. Mature C3 and C4 (MW of 190 and 204 kDa, respectively) consist of two (β-α) and three (β-α-γ) chains that form 13 domains: eight macroglobulin (MG1-8) domains, a linker, an anaphylatoxin-like (ANA) domain, a 'C1r/C1s, Uegf, Bmp1' (CUB) domain, a thioester-containing domain (TED) and a C3, C4 and C5 carboxy-terminal (C345C) domain[23–25]. Proteolytic removal of the small (9 kDa) ANA domain yields C3b and C4b, which expose their thioester

[1]Structural Biochemistry, Bijvoet Centre for Biomolecular Research, Dept. of Chemistry, Faculty of Science, Utrecht University, Utrecht, The Netherlands. [2]Present address: Thermo Fisher Scientific, Electron microscopy division, Eindhoven, The Netherlands. ✉e-mail: p.gros@uu.nl

moiety present in TED for covalent surface attachment[25,26]. C2 and factor B are five-domain serine proteases with MW of 92 and 90 kDa, respectively. A structure of pro-enzyme factor B shows that the pro-peptide region forms three 'complement-control protein' (CCP) domains and a helix αL in the linker region preceding its von Willebrand type A (VWA) domain. In factor B, pro-peptide helix αL blocks the conical positioning of C-terminal end of helix α7 of the VWA domain, which affects the orientation of the C-terminal serine protease (SP) domain[27]. A structure of pro-enzyme C2 is not available; however, structures are available for fragments C2a (CCP1-CCP2-CCP3) and C2b (VWA-SP)[28–30]. (We follow the recommendations of Bohlson et al.[31], referring to the small C2 fragment as C2a and the large fragment of C2 as C2b). Structural data on proconvertase and convertase formation from C4 and C2, forming C4b2 and C4b2b respectively, is limited to small-angle X-ray scattering data[32], but is available in detail for C3 and factor B. Crystal structures of the proconvertase C3bB and C3bB in complex with factor D reveal an 'open' conformation of factor B, with its scissile loop accessible for proteolytic activation by factor D[13]. However, small-angle X-ray scattering data[32] indicates that pro-convertase C4b2 is more like factor B in complex with cobra-venom factor (CVF), which displays a 'closed' conformation with an inaccessible scissile bond[33]. Furthermore, the crystal structures show an extensive binding interface between C3b and domains CCP2-3 of factor B, while interactions with the VWA domain of factor B involve binding of the C3b-carboxy terminus to the 'metal-ion dependent adhesion site' (MIDAS) of the VWA domain. A structure of the convertase C3bBb is only known in a dimeric complex with a *S. aureus* complement inhibitor (SCIN), (C3bBb-SCIN)$_2$[34]. In the absence of helix αL of the pro-peptide segment Ba, VWA- helix α7 rearranges and the SP domain re-orients in (C3bBb-SCIN)$_2$ by 142° compared to C3bB, while protease fragment Bb is connected to the C-terminal carboxylate moiety of C3b through the MIDAS in VWA. This contact point at the MIDAS is formed by a very slow on-off process, which is enabled by the extensive CCP2-3 contacts in C3bB[13]. Since the additional contacts provided by CCP2-3 are absent in Bb, reassociation of Bb with C3b is effectively impossible, hence explaining the irreversible dissociation[5,35]. By superimposing C3 onto C3b, the dimeric C3bBb-SCIN complex also serves as a model for the enzyme-substrate complex, C3bBb-C3. A dimeric interface formed by MG4-5 domains explains the inhibitory mode of compstatin[36]. However, this hypothetical model shows ca. 30-Å distance between the scissile loop in C3 and the active site in the Bb-SP domain[34]. A gap that might presumably be overcome by flexibility in the C345C-neck region in C3b, allowing a large swing of Bb towards the scissile loop.

Here, we present structures of human proconvertase (C4b2), convertase (C4b2b) and convertase-substrate (C4b2b-C3) complex from the classical and lectin pathways of complement activation derived from two cryo-EM single-particle experiments. Proconvertases were prepared by mixing C4b and C2. Convertases were generated by mixing C4b, a proteolytic inactive mutant S679A of C2 and an active C1s fragment in the presence of C3. Furthermore, to stabilize the convertase-C3 complex, we added a duo-nanobody with a long linker combining an anti-C3 and anti-C4b nanobody that do not inhibit classical and lectin pathway C3-convertase activity. Single-particle analysis of a first data set resulted in density maps of C4b2 at 3.5-Å resolution and analysis of a second data set yielded maps of C4b2b and C4b2b-C3 at 4.2 and 3.5-Å resolution, respectively.

## Results

### Proconvertase structure
4662 Images were collected of purified C4b2 complexes, resulting in a cryo-EM map at a 3.5-Å overall resolution (see Methods, Table 1 and Supplementary Figs. 1, 2). The reconstruction revealed well-resolved density for most domains, except for TED of C4b and SP of C2, which were not visible in the density; furthermore, C2-CCP1 displayed only partial density (Fig. 1a). To resolve structural heterogeneity and improve the representation of flexible regions, the dataset was further classified into two particle subsets, yielding maps at 4.0 Å (from 31,587 particles) and 4.2 Å resolution (18,613 particles) (Table 1, Fig. 1b and Supplementary Fig. 3a). In the 4.0-Å map, additional density corresponding to the SP domain became apparent, although at low local resolution (-5.5–7.5 Å). The 4.2-Å map revealed a very weak density for the SP domain, that was spatially displaced (Supplementary Fig. 3a); however, we omitted this domain from the model, because its orientation could not be established.

The conformation of C4b in the proconvertase structure was like that of the structure of C4b, except for a highly variable orientation of TED and a rotation by 29° of C345C, consistent with variations in C345C orientation observed in other structures of C4b (Supplementary Fig. 3b)[25,37,38].

The initial model for C2 was derived from previously published structures of C2a and C2b (previously reported as crystal structures of C2b and C2a, respectively)[28–30]. Using both the 3.5-Å and 4.0-Å maps, residues 209-233 of the CCP3-VWA linker were built; however, the scissile loop (residues 234–250) remained unresolved. VWA-helix α7 (res. 442-454) was remodelled and additional density in the 4.0-Å map allowed the building of the VWA-SP linker (res. 455– 462), and positioning of the SP domain (res. 463-752). Thus, the model of C2 in the proconvertase includes domains CCP1-3, linker, VWA and SP domain, but lacks the scissile loop (res. 234-249) and the C4b-TED domain.

CCP2-3 in C4b2 are arranged as in the C2a crystal structure[30], while CCP1 is rotated by 11°. The linker helix αL (res. 213-233) docks into a groove of the VWA domain and the C-terminal end of VWA-helix α7 (res. 451-454) appears dislocated, whilst the N-terminal part of helix α7 (res. 442-446) is positioned in its canonical groove, similar to that seen in the structures of factor B and C3bB[13,27]. The position of C2 SP in the 4.0-Å resolution map resembles that of the SP domain of C3bB in the open conformation (Fig. 1b, Supplementary Fig. 3c). Moreover, in C2, the P1 residue (Arg243) of the scissile bond is not bound by linker-helix αL and VWA-helix α7; and like in C3bB, the scissile loop is flexible and accessible in contrast to a bound and inaccessible Arg259 in free factor B and in CVF-B (Supplementary Fig. 3d)[33]. However, the position of the SP domain in C4b2 is highly variable, as indicated by the poor density; thus, suggesting a dynamic equilibrium, similar to what has been suggested for C3bB[39].

Contacts between C2 and C4b in the proconvertase are observed between the C2-CCP1 and C4b-C345C domains, between C2 CCP2-3 and C4b MG2, MG6-7 and CUB, and between C2 VWA and C4b C345C (Supplementary Fig. 4a). Extensive amino-acid interactions that are well defined by the density are observed at the C2 CCP2-3 site (Supplementary Fig. 5a). C2 CCP2 interacts with residues of the C4b (cleaved α) α' chain N-terminal region (α'NT) that lies over C4b MG7 and with residues of MG6, whereas several residues of C2 CCP3 interact with residues of C4b MG7 and CUB. A minor contact area is formed between C2 CCP1 and C4b C345C. Finally, the MIDAS region of C2 VWA contacts the C-terminal end of C4b C345C, where the C-terminus of C4b is expected to bind into the MIDAS. However, the density in this area is too poor for detailed interpretation (Supplementary Fig. 5b).

### Convertase structure
In one experiment, convertases and convertase-substrate complexes were formed, by mixing C3-C4b complexes linked by the duo-nanobody with C2 S679A and preactivated C1s in the presence of 2 mM Mg$^{2+}$ (see Methods). 4276 Images were collected from this sample yielding 17,073 particles. A density map at 4.2-Å resolution showed the convertase C4b2b with duo-nanobody segment NbC4bB12 attached (Fig. 2, Table 1, Supplementary Figs. 6–8). C4b and NbC4bB12 showed density consistent with the overall map resolution, however, the C2 domains VWA and especially the SP yielded significantly lower

**Table 1 | Cryo-EM data collection, refinement and validation statistics**

|  | C4b2 no SP PDB 9QJ5 EMDB-53199 | C4b2 PDB 9QJ4 EMDB-53198 | C4b2b PDB 9QPY EMDB-53288 | C4b2bC3 PDB 9QK2 EMDB-53217 |
|---|---|---|---|---|
| Microscope | Krios |  | Talos Arctica |  |
| Camera | Gatan K3 Summit + GIF |  | Gatan K2 Summit + GIF |  |
| Magnification | 105,000 |  | 130,000 |  |
| Voltage (kV) | 300 |  | 200 |  |
| Exposure time (s) | 2.2 |  | 4.0/6.0 |  |
| Number of frames | 50 |  | 40 |  |
| Electron exposure (e$^-$/ Å$^2$) | 50 |  | 50/52 |  |
| Defocus range (µm) | −1.0 to −2.2 |  | −0.8 to −2.6 |  |
| Pixel size (Å) | 0.836 |  | 1.04 |  |
| Micrographs (no.) | 4662 |  | 1700/2576 |  |
| Initial particle images (no.) | 765,438 |  | 589,457 |  |
| Final particles images (no.) | 111,565 | 31,587 | 17,073 | 177,801 |
| Map Resolution (Å) 0.143 FSC threshold | 3.5 | 4.0 | 4.2 | 3.5 |
| Refinement |  |  |  |  |
| Model Resolution (Å) 0.5 FSC threshold | 3.6 | 4.0 | 4.2 | 3.6 |
| Map sharpening B factor (Å$^2$) | −100 | −100 | −120 | −100 |
| Model composition |  |  |  |  |
| Non-hydrogen atoms | 12,950 | 15,254 | 16,917 | 29,917 |
| Protein residues | 1657 | 1948 | 2154 | 3793 |
| Ligands | NAG: 3, MG: 1 | NAG: 3, MG: 1 | NAG: 7, MG: 1 | NAG: 8, MG: 1 |
| R.m.s. deviations |  |  |  |  |
| Bond lengths (Å) | 0.003 | 0.004 | 0.003 | 0.003 |
| Bond angles (°) | 0.688 | 0.837 | 0.671 | 0.566 |
| Validation |  |  |  |  |
| MolProbity score | 1.70 | 1.86 | 1.89 | 1.58 |
| Clashscore | 8.48 | 11.32 | 14.88 | 4.95 |
| Rotamer outliers (%) | 0.42 | 0.54 | 0.05 | 1.13 |
| Ramachandran plot |  |  |  |  |
| Favored (%) | 96.40 | 95.80 | 96.71 | 95.93 |
| Allowed (%) | 3.60 | 4.20 | 3.24 | 4.07 |
| Outliers (%) | 0.00 | 0.00 | 0.05 | 0.00 |

local resolution (Fig. 2a). All domains of C2b and C4b and NbC4bB12 were rigid body fitted with minimal further adjustment and refinement (Fig. 2a), see "Methods".

The structure of convertase C4b2b revealed C4b in its common conformation[25,32], while the density for C2b was insufficient for detailed interpretation. Binding of the NbC4bB12 segment of the duo-nanobody to C4b was identical to what was observed previously[38]. Given the higher map resolution observed for the convertase-substrate C4b2b-C3 complex, structural changes from proconvertase C4b2 to convertase C4b2b will be discussed in detail later; instead here we describe the overall changes in domain positions only.

Removal of the pro-peptide segment C2a (CCP1-3 and αL) from the proconvertase C4b2 yields a series of displacements of C4b C345C, C2b VWA and C2b SP in C4b2b (Fig. 2b). In the absence of C2a, C2b VWA rotates by 34° with respect to the main body of C4b and 25° with respect to the C345C domain of C4b, while C4b C345C rotates by 22° (Supplementary Fig. 9a). In this new position, C2b VWA makes contacts with the CUB domain of C4b (Fig. 2b). This involves residues Arg1336 and Asn1337 of C4b CUB possibly interacting with residues Gln365 and Asn397 of C2b VWA, which were previously involved in interactions with C2 CCP3 in the proconvertase (Supplementary Fig. 9b). The removal of αL alters the VWA-SP interface, and the C2b SP domain

rotates by 124° (Supplementary Fig. 9c) relative to VWA. The overall result of these domain rotations is a shift of 33 Å (in S679A Cα position) of the SP active site and reorientation by 142° of the SP substrate-binding groove (Fig. 2b).

**Convertase-substrate complex**

Formation of C4b2b-C3 was facilitated by mutating the catalytic residue S679A of C2 and by stabilizing the convertase-substrate interaction through addition of anti-C3 hC3Nb3 and anti-C4b NbC4bB12 duo-nanobody. 177,801 particles, from the same data set as that used for the convertase structure, revealed C4b2b-C3 complexes at an overall resolution of 3.5 Å (Table 1, Supplementary Fig. 7, 8). C4b, C2b, C3 and anti-C4 NbC4bB12 were docked into the density. All domains were rigid-body fitted, followed by manual fitting and real- space refinement. The density was well resolved for all domains, except for C4b CUB and TED, C3 C345C and the C-terminal part of NbC4bB12 (Fig. 3a). Furthermore, no density was observed for the anti-C3 nanobody hC3Nb3, which binds to the top of the C345C of C3, consistent with its poorly defined density in negative-stain images of C3-hC3Nb3 complexes[40]. Binding of the anti-C4b NbC4bB12 was identical to what was observed in the convertase and as published previously[38]. Modelling anti-C3 hC3Nb3[40] onto the C345C yielded a 94-Å distance

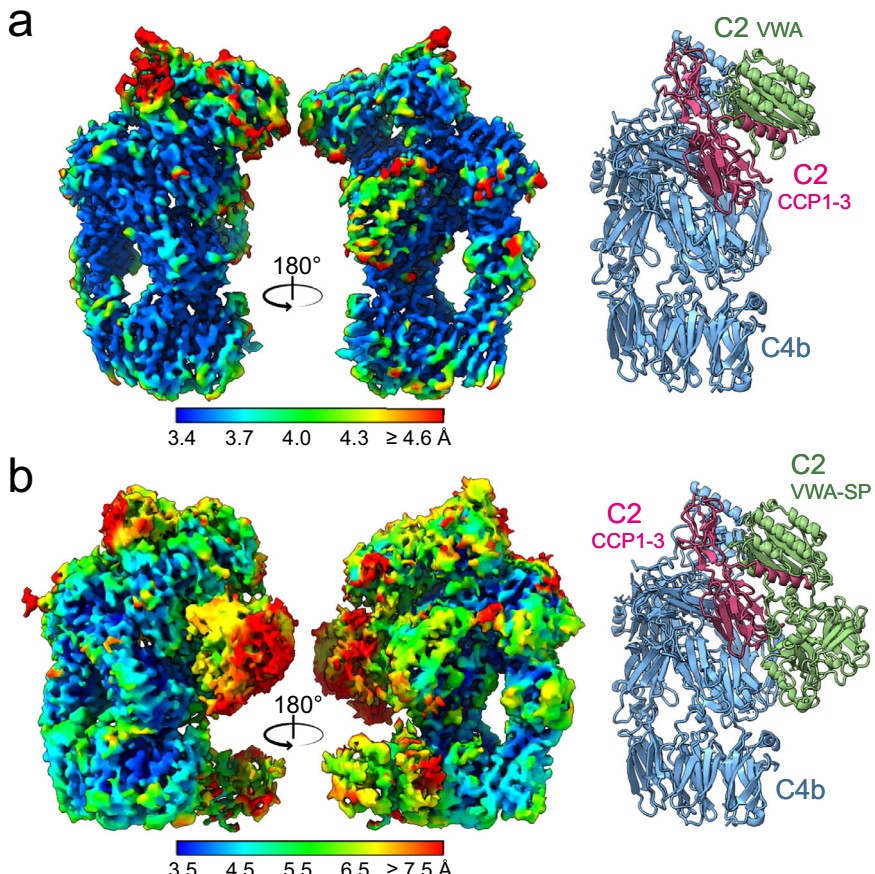

**Fig. 1 | Cryo-EM structures of the classical and lectin pathway proconvertase, C4b2. a** Cryo-EM density map at 3.5 Å overall resolution in two 180°-rotated orientations and cartoon-representation of the corresponding C4b2 structure. **b** Cryo-EM density map at 3.9 Å overall resolution obtained from a subset of particles and cartoon representation of the C4b2 model built in this map. Electron density maps are colored by local resolution from high (blue) to low (red), as indicated and models show C4b in blue, C2 CCP1-3 domains in purple and C2 VWA and SP domains in green.

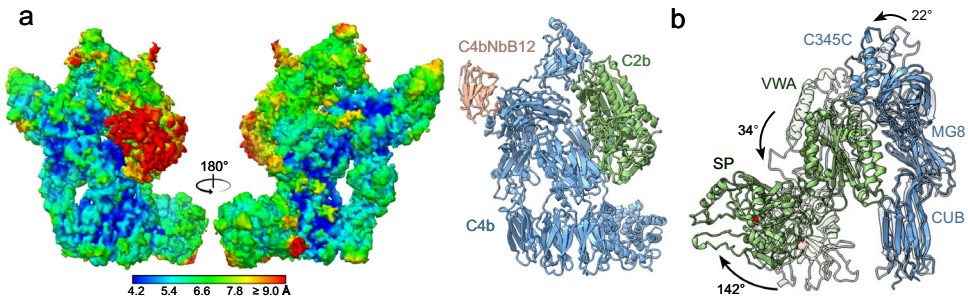

**Fig. 2 | Cryo-EM structure of the classical and lectin pathway convertase, C4b2b. a** Cryo-EM density map in two 180°-rotated orientations of C4b2b, colored by local resolution from high (blue) to low (red), and C4b2b model in cartoon representation with C4b in blue, C2b in green and duo-nanobody segment NbC4bB12 in light pink. **b** Domain reorientations during conversion from proconvertase to convertase. C4b2 and C4b2b were superposed on their C4b β chain. The figure shows C4b CUB, MG8 and C345C domains in blue and C2 or C2b VWA and SP domains in green, with C4b2 domains shown in faded colors. The arrows denote domain rotations going from C4b2 to C4b2b.

between N- and C-terminus of the nanobody models (Supplementary Fig. 10a), which was easily accommodated by the designed linker with an estimated length of 177 Å.

The density map of C4b2b-C3 showed the critical C2b-C4b contact of the convertase, between C2b VWA and C4b C345C, at a local resolution of 4–5 Å. The density is consistent with a carboxylate moiety of the C4b-γ chain chelating the $Mg^{2+}$ ion of the MIDAS in C2b VWA (Supplementary Fig. 10b). The orientations of C2b-VWA domains with respect to C4b C345C differ by up to 38° in the structures of C4b2,

C4b2b and C4b2b-C3 (Supplementary Fig. 10c). This variability suggests that the regular protein-protein interactions are weak and subsidiary to the pivotal interaction of the carboxylate moiety (chelating the $Mg^{2+}$) that sits at the end of a very short (2 residues) flexible C-terminal tail. The flexibility at this interface is consistent with the lower local resolutions observed at this interaction site.

The density map of C4b2b-C3 resolves VWA-helix α7 and the subsequent VWA-SP interface of C2b (Supplementary Fig. 10d, e). Upon removal of αL of the pro-peptide in C4b2, VWA-helix α7

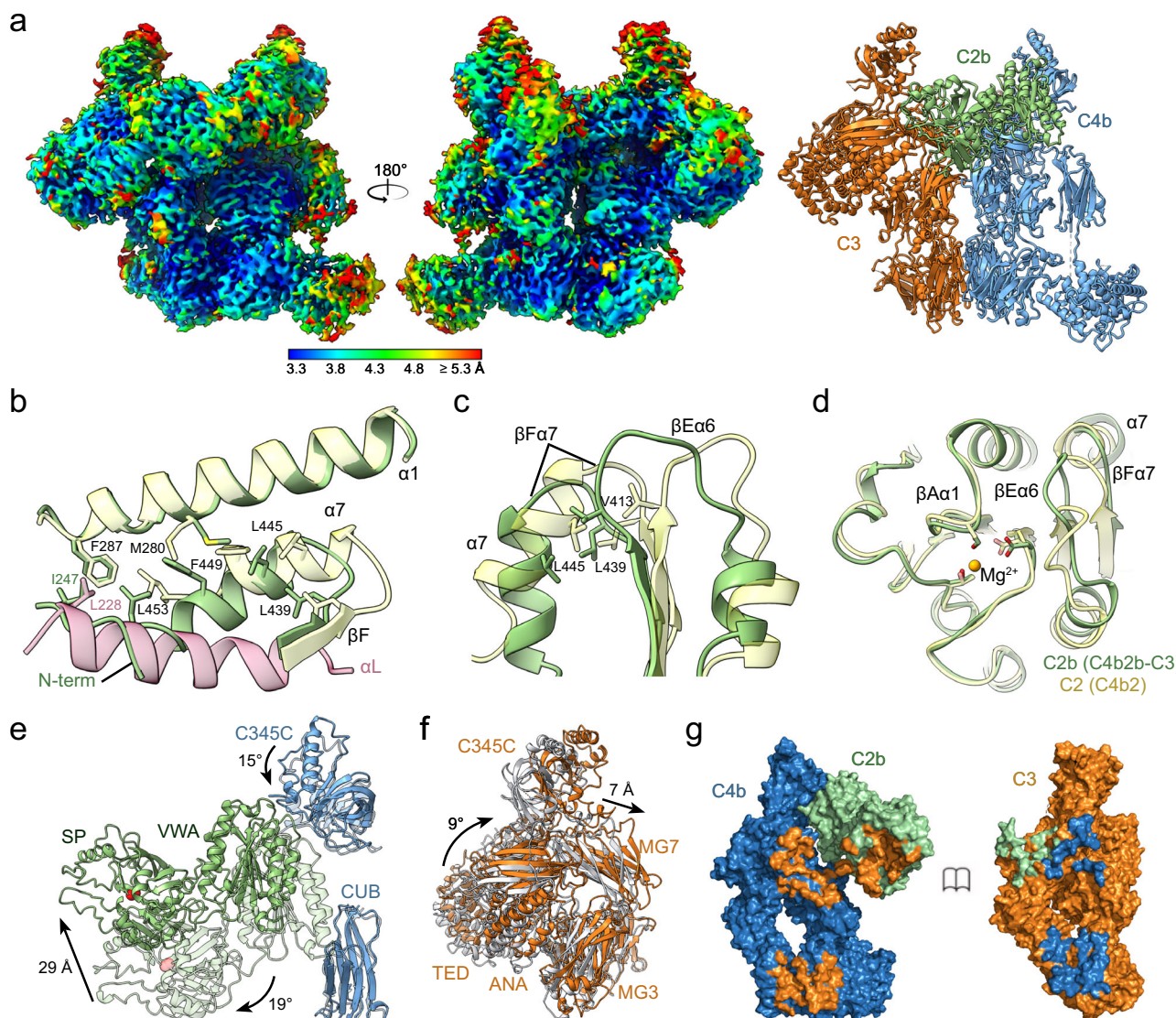

**Fig. 3 | Cryo-EM structure of the classical and lectin pathway convertase in complex with substrate C3. a** Cryo-EM density map in two 180°-rotated orientations of C4b2b-C3 colored by local resolution from high (blue) to low (red), and cartoon representation of the C4b2b-C3 model built in this map. C4b2b colors are as in Fig. 2 with C3 shown in orange. **b–d** Structural changes in the VWA domain upon conversion from C4b2 to C4b2b using the C4b2b-C3 structure. Helix αL of the C2 pro-peptide in C4b2 is shown in pink and VWA domains of C4b2b in green and C4b2 in light yellow. **e** Domain reorientations in C4b2b upon substrate C3 binding

with C3-bound and unbound convertases superposed using the C4b β-chain. C4b and C2b of C4b2b-C3 shown in blue and in green, respectively, with those of C4b2b in faded colors. Arrows denote the domain reorientations going from C4b2b to C4b2b-C3. **f** Domain reorientations in C3 upon binding to the convertase with C3 of C4b2b-C3 in orange and unbound C3 (PDB 2A73) in grey; molecules superposed using the MG1 and MG5 domains. **g** Surface representations of C4b2b (left) and C3 (right) from the C4b2b-C3 complex showing the contact sites. Proteins are colored as before with contact sites shown in the color of the opposing surface.

readjusts (residues 442-452), shifting by approximately one helical turn (Fig. 3b). Hydrophobic interactions between Val413 (loop βEα6), Leu439 (loop βFα7), and Leu445 (helix α7) are preserved, resulting in a concerted movement (Fig. 3c). This causes a displacement of loop βEα6 which lies in between α7 and the MIDAS site (Fig. 3d). Comparable changes occur in integrin-I domains[41,42], when forming a high affinity MIDAS state, with loops βAα1, βDα5 and βEα6 adopting equivalent conformations (Supplementary Fig. 10f). However, in (C3bBb-SCIN)$_2$ loop βFα7 of Bb (including Val430, equivalent to Leu439 in C2b) is affected due to contacts with SCIN, breaking the contacts between the three hydrophobic residues (Val430, Val455 and Val464) leaving the position of loop βEα6 almost unchanged in comparison to C3bB (Supplementary Fig. 10g). The C- terminal part of C2-VWA helix α7 (res. 451-454) moves into the position previously occupied by residues 221-226 of αL helix and now interact with the newly formed N-terminus of C2b (Fig. 3b). These changes in VWA affect the

linker 453-462 that dominates the VWA-SP interface (Supplementary Fig. 10e). The VWA-SP arrangement observed in the C4b2b-C3 complex is identical to that observed in structures of free C2b, indicating the stability of this VWA-SP arrangement.

Binding of C3 to the convertase C4b2b introduces changes in C4b2b and C3 (Fig. 3e, f). The C345C domain of C4b rotates 15° and both VWA and SP domain of C2b rotate by 19°. VWA and SP rotate as one by 13° with respect to C345C (Supplementary Fig. 10h). Overall, these motions shift the SP active site Cα of residue S679A by 29 Å towards the scissile loop of bound C3 (Fig. 3e). The structure of C3 changes by 9° in α-β chain orientation from that of free C3 (Fig. 3f)[23] to form two C4b-interaction sites that allows contacts with C2b and docking of the scissile loop in the protease active site (Fig. 3g).

One C4b2b-C3 contact site is formed by the MG4-5 domains of C3 and C4b (Fig. 4a); this interaction site is equivalent to the dimeric MG4-5 site observed between two C3b molecules in (C3bBb-SCIN)$_2$[34]

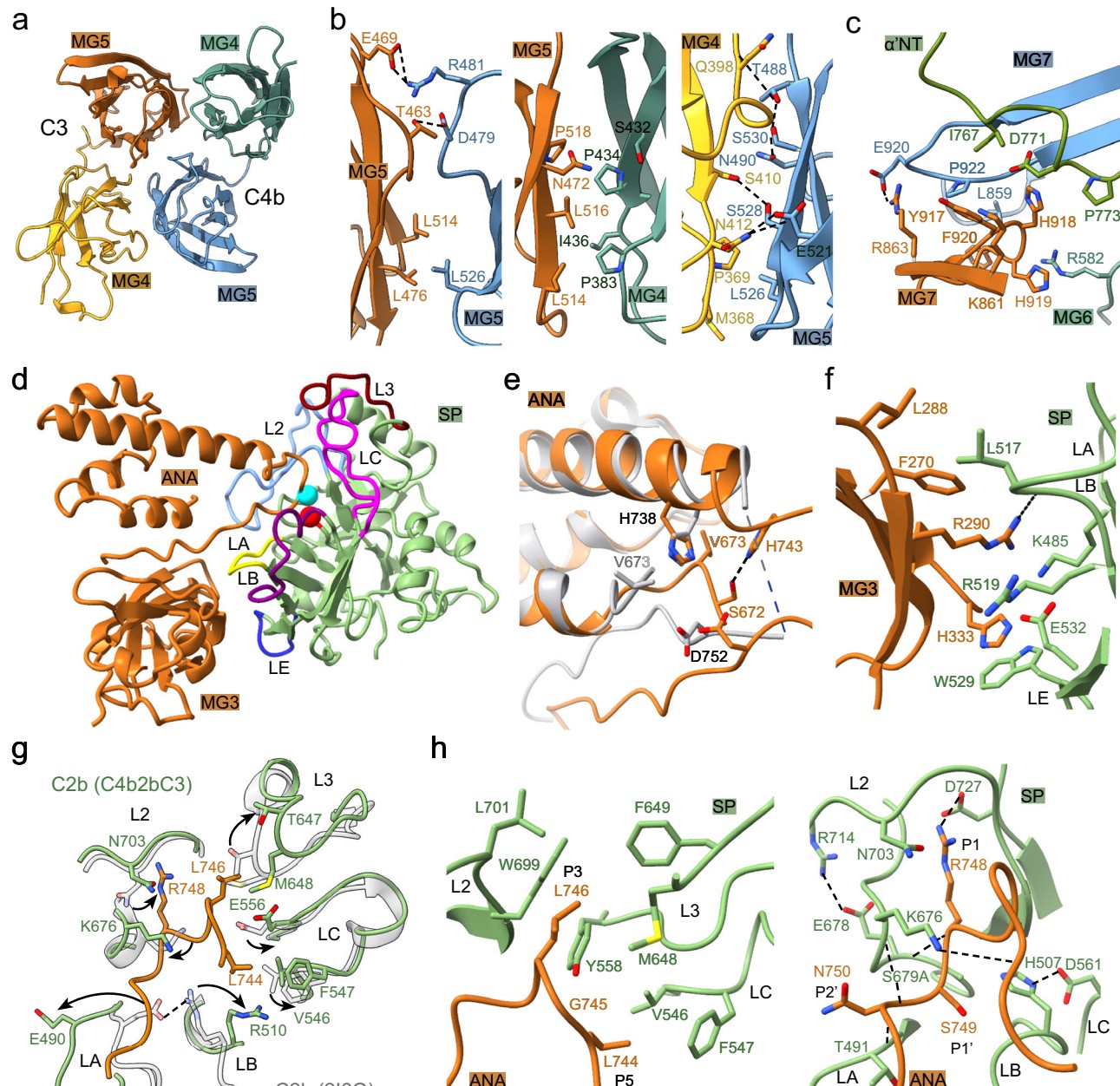

**Fig. 4 | Binding of C3 to convertase C4b2b. a** C4b2b-C3 contact site formed by C4b MG4 (green) and MG5 (blue) and C3 MG4 (yellow) and MG5 (orange). **b** Three views of the interactions between C3 MG4-5 and C4b MG4-5. **c** Interactions between the C3 MG7 and C4b MG6, MG7 and α'NT. **d** Binding of C3 domains MG3 and ANA (orange) to C2 SP (green). C2-SP loops that contact C3 are colored labelled "L" followed by their standard serine-protease names. C2 active-site serine and C3-scissile bond are indicated by red and cyan spheres, respectively. **e** Interactions between the modelled parts of the scissile loop in bound C3 (orange) compared to unbound C3 (grey). **f** Exosite interactions of C3 MG3 (orange) with C2 loop A, B and E (green). **g** Superposition of the SP active site of C2b (green) from C4bC2b-C3 compared to free C2b (grey; arrows indicate the changes from free C2b to C4b2b-C3. **h** Two views of interactions between C3-scissile loop (orange) and C2b SP (green).

(Supplementary Fig. 11a). The arrangement of MG4-5 of bound C3 differs by 11–12° from that of free C3, C3b and (SCIN-C3bBb)$_2$ (Supplementary Fig. 11b). The C4b-C3 MG4-5 interface buries 1309 Å$^2$ and involves a series of hydrophobic contacts supported by surrounding polar and charged interactions (Fig. 4b).

A second interface is dominated by the MG7 domains of C3 and C4b, with additional contacts provided by MG3 and MG6 of both molecules, covering 890 Å$^2$ (Supplementary Fig. 5c). A set of domain rearrangements in C3 allow formation of this second interface. MG3, 6 and 7 rotate by 33°, 5° and 15°, respectively (superposing unbound and convertase-bound C3 on their MG5 domains) (Supplementary Fig. 11c). In consequence, His918 at the tip of MG7 shifts 6.5 Å towards C4b,

where C3 loop 917-920 interacts extensively with C4b residues of MG6, MG7 and α'NT (Fig. 4c). Moreover, C3 MG6 forms electrostatic interactions with C4b-MG3 and MG6 and charged residues of C3 MG3 and MG7 provide additional interactions with C4b MG6 residues, further stabilizing the MG3, 6-7 contacts (Supplementary Fig. 11d).

Formation of the two C3-C4b interaction sites allow C3 domains MG3 and ANA (after 33° and 10° rotations, respectively, relative to MG5) to contact the C2b-SP domain (Supplementary Fig. 11c). Overall, the conformational changes in C3 allow the scissile loop to shift by 6 Å (in Cα position of Val741) and dock into the substate-binding groove (Fig. 4d). Internal H-bonds between His743-His738 and Ser672-Asp752 and interactions with substate-binding groove yield an ordered scissile

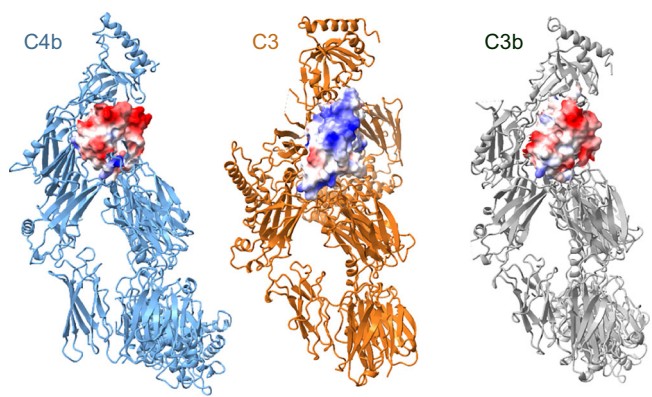

**Fig. 5 | Charge profiles of MG7 domains in convertase, substrate and product.** Electrostatic potential (negative charge in red and positive in blue) of MG7 of C4b (left) and C3 (middle) in C4b2b-C3 in open book orientation and C3b (right), taking α'NT bound to the surface of MG7 in C3b and C4b into account, showing that MG7-surface potentials of C4b-C3 are attractive and C4b-C3b are repulsive.

loop, whereas residues 742-750 were disordered in the model of free C3 (Fig. 4e). The MG3 domain of C3 interacts with residues of loop A (Lys485), loop B (Leu517, Arg519) and loop E (Trp529) of the C2b SP domain (Fig. 4f). Compared to free C2b (PDB 2I6Q)[28], loop A, B and 3 rearrange to allow access and binding of the scissile loop into the substrate binding groove (Fig. 4g); whereas, in a second structure of free C2b (PDB 2ODP)[29] loops A and B are disordered, while loop 3 resembles already the bound state (Supplementary Fig. 11e). In Bb of (C3bBb-SCIN)$_2$, the equivalent loops resemble the unbound state (Supplementary Fig. 11e). The protease, C4b2b, binds the scissile loop in the substrate-binding groove with Leu746, Arg748 and Ser749 docked into the S3, S1 and S1' substrate-binding pockets (Fig. 4h). The C2b catalytic site triad (His507, Asp561 and S679A) and oxyanion hole, stabilized by the guanidium group of Arg714, adopt arrangements compatible with catalytic activity, whereas the oxyanion hole is not fully formed in the structures of free C2b nor any of the Bb structures.

Overall, the structure of C4b2b-C3 resembles that of C3bBb-C3b taken from (C3bBb- SCIN)$_2$ (Supplementary Fig. 11f). Convertase C3b and C4b are arranged similarly. The position of protease fragment Bb lies in between that of C2b in C4b2b and C4b2b-C3, respectively (differing by rotations of 8° and 13°, Supplementary Fig. 11g), held in place by the two SCIN molecules bridging the dimeric pair of C3bBb complexes. As predicted previously, C3b-C3b interactions in the SCIN-inhibited C3bBb dimer mimic in part C3 binding to the convertase. However, C3b contacts the convertase (C3bBb) only through the MG4-5 domains, while the two C3b-MG7 domains are ca. 14 Å apart in (C3bBb-SCIN)$_2$. In comparison, C3 MG7 of C4b2b-C3 and the corresponding C3b MG7 of (C3bBb-SCIN)$_2$ differ by 40° in orientation. Moreover, C3b MG7 binds the negatively charged α'NT tail onto its surface. Altogether, while in C4bC2b-C3 a positively charged surface of C3 MG7 faces a negatively charged surface of C4b MG7 and α'NT, in (C3bBb-SCIN)$_2$ two negatively charged surfaces of MG7 and α'NT oppose each other (Fig. 5).

## Discussion

We presented the cryo-EM structures of C3 proconvertase (C4b2), C3 convertase (C4b2b) and C3 convertase-substrate (C4b2b-C3) complexes, as they occur in the classical and lectin pathways of complement activation. Following the recommendations of Bohlson et al.[31], we refer to the large fragment of C2 as C2b, and not C2a (which originally indicated 'activated' fragment of C2). Hence, the fragments C2a and Ba vs. C2b and Bb of homologous proteases C2 and factor B refer to the equivalent parts. The structures of C4b2, C4b2b and C3-C4b2b, together with previously reported structures of the alternative

pathway, proconvertase C3bB, C3bBD and SCIN-inhibited convertase (C3bBb-SCIN)$_2$, provide a comprehensive view of the structural steps underlying C3 convertase formation, stability and activity.

Formation of the bimolecular convertase complexes requires the pro-enzymes factor B and C2 to provide extensive interactions through the CCP2-3 domains of their pro-peptide segments to bind C3b or C4b, respectively. Binding through the pro-peptide segments, allow the C-termini of C3b and C4b to chelate the MIDAS-Mg$^{2+}$ ion present in the VWA domain of factor B and C2. We observed (up to 38°) variations in relative orientation of the C2 and C2b VWA domains to C4b C345C. This flexibility indicates that interactions outside the MIDAS in C4b2b are weak and underscores that chelation of the C3b or C4b carboxy terminus to the Mg$^{2+}$ ion is pivotal for the bimolecular complexes. A common arrangement for VWA-loops βAα1 and βDα5 around the MIDAS is observed in the proconvertase structures C4b2 and C3bB, consistent with ligand-bound state of the MIDAS[42]. Proteolytic activation (from proconvertase to convertase) repositions VWA-helix α7, loop βFα7 and loop βEα6, which lies adjacent to the MIDAS. Possibly, this rearrangement yields a transition of low-to-high ligand-binding affinity at the MIDAS, similar to that of integrin I domains. This would explain an increased resistance of the convertase, compared to the proconvertase, to EDTA and high salt[32,43]. Overall, while insufficient interactions outside the MIDAS prohibit the enzyme fragments Bb or C2b to reassociate with C3b or C4b, respectively, a high-affinity binding state of the MIDAS in convertases would help to explain the very slow dissociation, which together explains the ca. 1–1.5 min half-life times of the irreversible dissociating C3 convertases.

The catalytic C2-SP domain is oriented highly variably in the C4b2 proconvertase complex, with its most occurring positioning observed in a 'downward' orientation, close to C4b MG2, MG6 and CUB. Moreover, the scissile bond residue Arg243 is not sequestered by the VWA in C4b2 but is disordered and exposed. A similar 'downward' SP orientation and exposed arginine is observed in the structure of C3bB that is compatible with proteolytic activation by factor D. Thus, our data do not support an additional activation step required by the initiation complexes C1 and MBL to expose the C2- scissile bond in C4b2, as hypothesized[32]. Cleavage of proconvertase C2 radically alters the orientation of the SP domain of the convertase complex by 142°. Upon removal of pro-peptide helix αL, the C-terminal end of VWA-helix α7 readjusts and the VWA-SP changes drastically. The resulting VWA-SP conformation in the convertase C4b2b is identical to that of free C2b, indicating that this conformation is stable. Upon substrate binding, the C2b segment of C4b2b swings, moving the catalytic site over 29-Å distance. C2b-SP loops A, B and 3 rearrange to allow access of the substrate scissile loop into the substrate-binding groove of C2b. Similar changes would be required to open the SP-loops A, B and 3 in C3bBb. Surprisingly, the two free-C2b crystal structures differ from each other with respect to loops A, B and 3; with one like the unbound state[28] and the other one, largely disordered, reflecting a transition state[29]. Besides C2b-SP interactions with the scissile loop, additional interactions are observed between domain MG3 and loop E of C2b SP. The catalytic triad and oxyanion-hole are fully formed in the structure of convertase-substrate complex, whereas in all other structures the oxyanion hole is not completely formed.

Binding of the substrate C3 to C4b2b is mediated by two C3-C4b binding sites. One site is formed by hetero-dimeric interactions of MG4-5 domains of C4b and substrate C3. Homo-dimeric interactions were previously found at the same MG4-5 site between the two C3b molecules in (C3bBb-SCIN)$_2$. Interactions at this site are disrupted by members of the compstatin family that bind C3 at the MG4-5 surface[44]. An intricate set of domain adjustments occur within C3 upon binding C4b2b, with rotations of 33°, 10° and 15° for MG3, ANA and MG7, respectively, and overall change of 9° in α-γ to β chain orientation. These changes accommodate formation of a second interaction site with C4b formed by MG3, 6 and 7, as well as the α'NT region. The

inhibitor CRIg from Kupffer cells binds domain MG3 and 6 and, thereby, blocks these interactions directly[45]. The interactions with C3 MG7 explain why a C3-deletion mutant Δ923-924[46] cannot serve as substrate to C3 convertases but could still be active as a C3b ligand in a C3 convertase[46]. Formation of this second C3-C4b binding site and accompanying conformational changes in C3 move the scissile loop by 7 Å and allow docking of the C3-scissile loop into the active site of C4b2b.

Finally, structural comparison of the convertase-substrate, C4b2b-C3, complex with the SCIN- inhibited dimer (C3bBb-SCIN)$_2$ suggests a potential substrate-release mechanism, using (C3bBb-SCIN)$_2$ as a model of a stabilized convertase-product (C3bBb-C3b) complex[34]. Whereas the substrate C3 binds the convertase in C4b2b-C3 through two binding sites on C4b, the product C3b interacts through only one interface, the MG4-5 site, in the stabilized C3bBb-C3b complex. Upon cleavage of C3 into C3b, the product undergoes drastic conformational changes, in particular within the α chain, with large domain reorientations and a relocation of the newly created α′NT that binds into a surface groove of the (rotated) MG7[26]. In the uncleaved C3(H$_2$O) similar conformational changes occur, where relocation of the ANA domain through a transiently enlarged MG1-6 ring places the identical residues (of the α′NT) in the same position on MG7[47]. In C3b and C3(H$_2$O), the β chain MG4-5 domains remain available for interactions, forming dimeric interfaces in C3b-C3b[36] and C3(H$_2$O)-C3(H$_2$O)[47], which are similar to the hetero-dimeric MG4-5 interface in C4b2b- C3. However, in its new orientation and with α′NT bound, the product C3b MG7 presents a negatively charged surface that opposes a similar negatively charged surface of the convertase C4b, C3b or C3(H$_2$O). In contrast, in C3 the presented surface of MG7 is positively charged and suitable for binding to MG7 of C4b, C3b or C3(H$_2$O) of a convertase. Therefore, MG7 domains of the convertase-substrate complexes are electrostatically attractive, whereas these domains of the convertase-product complexes are repulsive. Thus, we hypothesize that this charge reversal of substrate into product is part of the product-release step. Product release by the C3 convertases is possibly a determining step that distinguishes between continued surface opsonization by producing C3b molecules and initiation of the terminal pathway by switching to C5 convertase activity.

## Methods

### C4b2 complex formation and purification
Similar to Mortensen et al.[25,32], we used C2 and C4b derived from pooled human serum (thus containing a mixture of C4A and C4B isotypes) (Complement Technology, Inc.) Both proteins were mixed at final concentrations of 3.2 and 2.1 μM, respectively, in 20 mM HEPES pH 7.4, 150 mM NaCl, 3 mM MgCl$_2$, supplemented with 0.2% glutaraldehyde for crosslinking. The mixture was incubated for 30 min. on ice and quenched with 1 M Tris pH 8.0 to a final concentration of 100 mM. The C4b2 complex was purified via size exclusion chromatography (SEC) on a Superdex 200 Increase 10/300 column pre-equilibrated in PBS and kept on ice until cryo-EM grid preparation.

### C4b2b-C3 complex formation and purification
C3 was purified from human serum by differential polyethylene glycol precipitation to enrich complement components, followed by anion-exchange chromatography on a DEAE matrix to isolate native C3. Final polishing steps removed residual serum proteins and yielded homogeneous, functional C3[23]. Previously generated constructs for inactive C2 (S679A) and a C1s fragment consisting of domains CCP1, CCP2 and the SP domain (res. 293-688) with the native cleavage site at residues 433-438 substituted with an enterokinase cleavage site (DDDDK) were used. N-terminal hexahistidine tagged C2(S679A) and C1s, were transiently expressed in HEK293-E+ and HEK293-ES cells, respectively

(Immunoprecise Antibodies BV) and harvested 6 days post-transfection. A duo-nanobody construct was engineered by fusing anti-C4b[38] and anti-C3 nanobody hC3Nb3[40] separated by a linker containing a non-complement specific nanobody 4C8[48] flanked by thrombin cleavage sites followed by 3xGSS repeats. The construct was cloned into a modified pET13 vector encoding a C-terminal hexahistidine tag and expressed in *E. coli* BL21(DE3) pLysS. Hexahistidine-tagged proteins were purified using Ni-Sepharose for the duo-nanobody and Ni-Sepharose Excel beads for C2 (S679A) and C1s (DDDDK) (GE Healthcare) at 4 °C, using standard protocols and a further SEC step using a Superdex 75 10/300 column (GE Healthcare) pre-equilibrated in 20 mM HEPES pH 7.4, 150 mM NaCl. Peak protein fractions were pooled, analyzed by SDS-PAGE gel to confirm the presence of all protein components, and stored at 4 °C for up to 3 days prior cryo-EM grid preparation.

25 μg of C1s (DDDDK) at an initial concentration of 28 μM was activated by cleavage with 16 units enterokinase (New England Biolabs Inc, P8070L) at 37 °C for 1 h in a total volume of 21 μL in 20 mM HEPES pH 7.4, 150 mM NaCl buffer containing 2 mM CaCl$_2$. Human serum-derived C4b (Complement Technology, Inc), C3, and the duo-nanobody were mixed to a final concentration of 5 μM each. The C4b-C3 duo-nanobody complex was purified by SEC using a Superdex 200 Increase 3.2/300 column pre-equilibrated in 20 mM HEPES pH 7.4, 150 mM NaCl, 2 mM MgCl$_2$. The C4b-C3 duo-nanobody complex (0.20 mg/ml), C2 (S679A) (0.50 mg/ml) and activated C1s(DDDDK) (0.05 mg/ml) were mixed in a 25:5:1 volume ratio, incubated for 30 sec at room temperature and immediately used for cryo-EM grid preparation.

### Cryo-EM grid preparation and data collection
3 μl of C4b2 (0.20 mg/ml) or C4b2b-C3 (0.23 mg/ml), were applied to a glow discharged R1.2/1.3 200 mesh Au holey carbon grid (Quantifoil) and plunge-frozen in liquid ethane using a Vitrobot Mark IV (Thermo Fisher Scientific) at 4 °C and 22 °C, respectively.

C4b2 cryo-EM data were collected on a 300 kV Titan Krios transmission electron microscope (Thermo Fisher Scientific) equipped with a Gatan K3 Summit direct electron detector and a Gatan BioQuantum post-column energy filter operated with a 20 eV slit width. Data were acquired in counted super-resolution mode using EPU-2.10.00 (Thermo Fisher Scientific) at a nominal magnification of 105,000×, corresponding to a super-resolution pixel size of 0.418 Å/pixel. 4662 movies were collected using a defocus range of −1.0 to −2.2 μm, at a total dose of 50 e⁻/Å² across 50 frames with 2.2 s exposure time.

C4b2b-C3 cryo-EM data were acquired using a 200 kV Talos Arctica transmission electron microscope (Thermo Fisher Scientific) equipped with a K2 Summit direct electron detector (Gatan) and a post-column GIF Quantum energy filter (Gatan) operating with a 20 eV slit width. Two datasets were collected from the same specimen in counting mode using EPU-2.9.00 (Thermo Fisher Scientific) at a nominal magnification of 130,000×, corresponding to a calibrated pixel size of 1.04 Å at the specimen level. A total of 1700 and 2576 movies were collected for the two datasets, respectively, over a defocus range of −0.8 to −2.6 μm. Each movie consisted of 40 frames. The first dataset was recorded with a total dose of 52 e⁻/Å² and a total exposure time of 4.0 seconds, and the second with a total dose of 50 e⁻/Å² over 6.0 seconds.

### Processing of C4b2 cryo-EM data
Cryo-EM data for C4b2 were processed using Cryosparc v3.2/3[49]. The dataset was collected in super resolution mode (0.418 Å/pixel), it was imported and 2× binned to a pixel size of 0.836 Å/pixel, followed by patch- based motion correction, patch-based CTF estimation, and movie curation. After curation, 4486 movies were retained for further processing. Initial blob picking was performed on all curated micrographs, followed by particle extraction using a box size of 480 pixels,

binned to 256 pixels. Three iterative rounds of 2D classification and particle selection yielded 765,438 particles. These particles were subjected to ab initio reconstruction to generate three initial models (one high-quality model and two decoys). The resulting models were used in several rounds of heterogeneous refinement, followed by non-uniform refinement, yielding an improved 3D reconstruction from 190,374 particles. To improve the resolution of the VWA and SP domains, a mask covering C2 and the C4b-C345C domain was applied to the refined volume, and the data were subjected to 3D variability analysis, producing four distinct clusters. These clusters underwent heterogeneous refinement, and the most well-resolved cluster contained 111,565 particles. These particles were re-extracted at the original box size (0.836 Å/pixel), and subjected to non-uniform refinement, producing a 3.5-Å map, according to the gold standard FSC = 0.143 criterion. This map showed no density for the C4b-TED and the C2-SP domain, and only partial density for C4b CCP1.

Further 3D-variability analysis and iterative heterogeneous refinement of the 111,565 particles resulted in two maps that revealed the C2-SP domain in distinct orientations. One cluster, composed of 30,885 particles, yielded the best-defined features of C2. The second, from 18,613 particles, produced a 4.2-Å map with weak density for the SP domain. The best-defined C4b2 map was then reprocessed through multiple rounds of 3D-variability analysis and heterogeneous refinement, starting again from the 190,374 particles. The final non-uniform refinement yielded a 4.0-Å resolution map, based on 31,587 particles. This reconstruction revealed the C2-SP domain at a local resolution of ~5.5–7.5 Å (see Table 1).

### Processing of C4b2b-C3 data

Two datasets of the C4b2b-C3 complex were processed in CryoSPARC v4.2.0/4 without binning. Each dataset underwent patch-based motion correction and CTF estimation, followed by movie curation, resulting in the selection of 1384 and 1906 movies, respectively. An initial subset of 100 movies was used for blob picking, generating 2D templates that guided particle picking across the full datasets. A total of 448,154 and 615,097 particles were extracted using a box size of 380 pixels, Fourier-cropped to 256 pixels, and subjected to multiple rounds of independent 2D classification. Selected 2D classes from both datasets were merged and submitted to an additional round of 2D classification and selection, yielding a final pool of 589,457 particles. These particles underwent multiple rounds of heterogeneous refinement using decoy models, followed by non-uniform refinement, resulting in a reconstruction of the C4b2b-C3 complex at 3.5 Å resolution (gold-standard FSC = 0.143) from 177,801 particles. These were re-extracted at the original box size prior to the final refinement.

One of the decoy models revealed clear features of C4b along with partially resolved density for C2b, corresponding to the C4b2b complex. This map was further processed following the same strategy used previously to resolve the SP domain in C4b2. A mask covering the C4b-C345C domain and C2b was created, and the map along with its corresponding particles, was subjected to multiple rounds of 3D variability analysis and iterative heterogeneous refinement, ultimately improving the features of C2b. From this subset, 17,073 particles were re-extracted and refined, yielding a 4.2 Å resolution map.

### Model building and refinement

The C4b2 model was generated by docking the crystal structures of C4b (PDB 5JTW)[50], C2b (PDB 2I6Q)[28], and C2a (PDB 3ERB)[30] into the 3.5 Å cryo-EM density map using domain-by-domain fitting in ChimeraX[51]. Rigid-body refinement was performed in Phenix[52], followed by manual rebuilding in regions where density deviated from the initial models. Iterative cycles of model building and local refinement were conducted in Coot[53] and Phenix. Several regions were omitted from the final model due to insufficient density, including the C4b-TED domain (residues 985–1325), residue 757 of the C4b

N-terminal α-chain, residues 1391–1392 linking C4b-CUB to MG8, and the C2- SP domain (residues 455–752). The C2-CCP1 domain was refined by rigid-body fitting only, with residues 74–80 excluded due to poor density. Density allowed modeling of the first N-glycan at C4b residues N226, N862, and N1328, as well as the modelling of C2 helix αL (residues 209–229) and rebuilding of C2 helix α7 (residues 444–454). The scissile loop (residues 230–250) was unresolved and therefore not built. The C-terminal tail of the C4b α'-chain (residues 1414–1421), absent in the original crystal structure, was resolved and built de novo. The final model was refined in real space using Phenix with standard geometric restraints.

For the 3.9 Å cryo-EM map, which contained additional density for the C2-SP domain (residues 463–752), the refined C4b2 model served as a starting point. The SP domain (PDB 2I6Q)[28], without glycans, was rigid body fitted in ChimeraX and refined in Phenix. The linker between the C2-VWA and C2-SP domains (residues 455–462) was manually built in Coot. Density supported modeling of residues 231–233 in the scissile loop, while residues 234–250 remained unresolved. Refinement followed the same protocol as above, with the C2-CCP1 and C2-SP domains kept fixed after rigid-body fitting. This yielded a lower-resolution C4b2 model incorporating the SP domain.

To generate the C4b2b model, the structure of C4b in complex with NbC4bB12 (PDB 7B2Q)[38] was docked into the 4.2 Å cryo-EM map. As this model lacked the C4b-C345C domain, residues 1594–1744 from the C4b crystal structure (PDB 5JTW) were used. Given the relatively low resolution of the map, all C4b domains, NbC4bB12, and C2b (PDB 2I6Q) were rigid body fitted in ChimeraX and refined in Phenix. Limited manual building was performed in Coot, including extension of the C4b N-terminal α'-chain (residues 762–767) and repositioning of sidechains of residues Arg1336 and Asn1337. Residue Ser679 in C2b was mutated to alanine, reflecting the construct used for sample preparation. The first N-glycan was retained or added at glycosylation sites N226, N862, N1328, and N1391 of C4b, and at N290, N467, and N470 of C2b based on map density.

The C4b2b-C3 model was built using the same approach as for C4b2b, using the C2b structure (PDB 2ODP)[29] and correcting point mutations according to the experimental construct (A261C reverted to wild-type and S679A introduced). C3 (PDB 2A73)[23] was also incorporated into the model. Initial domain-by-domain rigid-body fitting was followed by iterative manual rebuilding in Coot and real-space refinement in Phenix with geometric restraints. The 3.5 Å map allowed modeling of protein C3 N-terminal α-chain residue 672 and scissile loop residues 742–750. Loop regions in A, B and 2 of C2b (residues 487–490, 509–517, and 707–711, respectively) were built. N-glycans were modeled at C2b residues N290, N467, N471, and N621, and at residues N226 and N862 for C4b. For protein C3, residues 74–75 and 312–313 were modeled, while residues 102 and 1497–1499 were omitted due to lack of density. The first N-glycan was also added at residues N85 and N939 where density permitted it and refined as previously described.

Illustrations portraying protein structures were prepared using Pymol (Schrödinger) and UCSF ChimeraX[51].

### Ethics statement

The work presented in this study complies with all relevant ethical regulations. The use of human blood from healthy volunteers was approved by the Medical Ethics Committee of the University Medical Center Utrecht (METC protocol 07-125/C approved on March 1, 2010). Donors provided written, informed consent in accordance with the Declaration of Helsinki.

### Reporting summary

Further information on research design is available in the Nature Portfolio Reporting Summary linked to this article.

## Data availability

The model coordinates and cryo-EM density maps of the structures have been deposited in the Protein Data Bank (PDB) and the Electron Microscopy Data Bank (EMDB) under the following accession numbers; PDB 9QJ5 and EMD-53199 for C4b2 (no SP), PDB 9QJ4 and EMD-53198 for C4b2, PDB 9QPY and EMD-53288 for C4b2b, and PDB 9QK2 and EMD-53217 for the C4b2b-C3 complex. The EMDB depositions include unfiltered-half maps, non-sharpened unmasked maps and sharpened masked maps. The source data underlying Supplementary Figs. 1b-c, 6b and 7a, c are provided as a Source Data file. As described in the Methods section, structural models used to initiate model building are Protein Data Bank entries 5JTW, 2I6Q, 3ERB, 2ODP, 7B2Q and 2A73. Source data are provided with this paper.

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

## Acknowledgements

We acknowledge Linda Markus for her work in the initial stages of the project, Joke Granneman for her help in cloning, Nicoleta Ploscariu for insightful conversations on the project and Willem Noteborn, Netherlands Centre for Electron Nanoscopy (NeCEN), and Jamie Depelteau for their help and assistance during data collection. Additionally, we thank the support staff Stuart Howes and Chris Schneijdenberg of the cryo-EM facility at Utrecht University. This work was supported by Consejo Nacional de Ciencia y Tecnología in Mexico (CVU 604718) (K.I.O.B). This project has received funding from the European Research Council (ERC) under the European Union's Horizon 2020 Framework Programme Grant Agreement No. 787241 (P.G.), and by the Dutch Research Council (NWO) (Project No. 01.80.104.00) (P.G.).

## Author contributions

I.S.M. contributed at initial stages of the proconvertase project and protein purification. K.I.O.B. carried out design of linker constructs, protein purification, biochemical assays, electron microscopy sample preparation, data processing, model building and model refinement. K.I.O.B., T.H.C.B. and P.G. interpreted the data. P. G. supervised the project. K.I.O.B., T.H.C.B. and P.G. wrote the manuscript with critical input from I.S.M.

## Competing interests

The authors declare no competing interests.
