## [Transparent Peer Review file · Nature Communications]

Structural insights into C3 convertase activity of the classical pathway of complement

Corresponding Author: Professor Piet Gros

Version 0:

Reviewer comments:

Reviewer #1

(Remarks to the Author)

The is an important manuscript that reports the Cryo-EM structures of the complement classical / lectin pathway C3 proconvertase, C3 convertase and C3 convertase with its substrate C3. These structures provide significant information about the structural or conformational changes of the activated components during the complement activation processes. That information would be relevant for drug designs of medical intervention controlling immune effector functions and ameliorating inflammatory response to reduce complement-mediated tissue injuries. There are two important issues that are desirable to be addressed.

First, the source of C4 or C4b used was only vaguely described (line 380). Were "serum-derived C4b and C2 originated from a single human subject or from mixed sera of multiple human subjects. This is particularly relevant for human C4 because of its extensive polymorphism of C4A and C4B isotypes with multiple allotypes in each protein. The polymorphic sites in C4 are overwhelmingly clustered at the TED domain and most human subjects consist of both acidic and basic isotypes. Unfortunately, cryoEM structures of the TED domain were not resolvable in this manuscript and thus their structural and functional relevance were obscure. In retrospect, the use of source proteins from single human subjects with minimal variations would have yielded more definitive or informative structures.

Second, dis-similar to many colleagues including the same senior author of this manuscript who published on complement structures, the heavy chain of C2 with serine proteinase and vWA domains (after activation⁰ was referred to as C2b instead of C2a. This creates unnecessary confusions, and feelings to many colleagues. Irrespective to the reasons of the nomenclature switch, it would be appropriate to add a footnote explaining the change of nomenclatures, or use a new set of names for the heavy and light chains of activated C2. The authors could acknowledge the limitations of using the reversed names for C2a and C2b in the Discussion

Reviewer #2

(Remarks to the Author)

The paper by De la O Becerra et al presents Cryo EM structures of the classical C3 proconvertase, convertase and the convertase-substrate complex. A number of novel insights are provided as to the similarity of the classical convertase to the previously described alternative convertase as well as the mechanism of classical convertase activity and formation. The structure of the convertase-C3 complex is very interesting and resolves questions related to how interaction of the protease domain with the scissile bond is achieved. Previous work with the C3bBb-SCIN complex suggested a general explanation of the mode of C3 binding, especially the importance of the MG4-5 domains of both components but also indicated that there must be more to the story. The structure of the classical convertase-C3 structure answers the remaining questions showing that a number of relatively small structural movements combine create additional converatse-C3 interactions to form a clear enzyme-substrate complex. Apparently, the absence of the SCIN inhibitor also allowed explanation of the increased strength of the VWA MIDAS-C4b (and presumably C3b) interaction after activation of the protease component. The structures in the paper are of high quality with poorly resolved regions being restricted to areas of less relevance to the study.

The complex with the substrate produced a very good structure. The scissile bond loop is very well resolved as is the newly described contact site involving the MG7, MG6 and MG3 domains. The chromatogram in figure S6 seems to suggest that the complex was probably stable enough to be analyzed without the linker peptide.

The paper is well written and the figures are of good quality. Overall, I don't think the paper requires any real modification.

Reviewer #3

(Remarks to the Author)

This interesting manuscript tackles one of the unsolved structural problems in the complement field, the assembly of the classical pathway C3 convertase. As such it is of some importance - particularly in the context of evolving therapeutic approaches to inhibiting complement activation.

One general concern is the decision to call the convertase C4b2b. The convertase was, from its first description in the 1960s, termed C4b2a. In recent years there has been a move from some quarters to "rationalise" by referring to the convertase as C4b2b; as the authors know, this is an area of continued disagreement in the field. This should be noted in the Introduction and the reason for deciding on C4b2b in the current Ms stated.

The authors are very expert complement structural biologists and I cannot criticise their cryoEM analyses and interpretation; however, I do have some concerns that should be addressed.

My major concern is around the generation of the complexes for analysis.

The C4b2 complex is generated by simply mixing commercially sourced plasma-derived C4b and C2 at 2.1 and 3.2 μ m respectively in the presence of glutaraldehyde to non-specifically cross-link. How was this method arrived at and what is the evidence that the complex generated in this artificial way represents the labile proconvertase? Indeed, the gels shown in supp Fig 1 suggest that there is very little C2 in the isolated final complex!

Conversion of proconvertase to convertase is mediated (I think - not clear in methods) by incubation of proconvertase with "preactivated C1s" but evidence of cleavage is not provided.

To generate convertase-substrate complexes, C3-C4b complexes are first generated by mixing the two proteins together with a cross-linking nanobody. Again, what is the evidence that this artificially generated complex adopts a conformation relevant to the natural convertase? This complex is then incubated with inactivated C2 and preactivated C1s to yield a nanobody-stabilised C4b2bC3 complex. once more, what is the evidence that this complex is representative of the native complex? supplementary figure 6 shows gels of the C3-C4b-nb complex but not of the complex incorporating C2b, essential for the characterisation.

Minor points:

Figure legends are brief and would benefit from more explanation for non-experts;

Some annoying abbreviations - for example, line 68 and elsewhere - "resp." for respectively, should be eliminated.

Version 1:

Reviewer comments:

Reviewer #1

(Remarks to the Author)

The authors addressed the questions raised by this reviewer. It's the authors' choice to use the revised nomenclatures of C2a and C2b. On another issue about the protein polymorphisms of C4A and C4B, it was unfortunate that the authors ignored the phenomenon.

Reviewer #3

(Remarks to the Author)

This is an important paper and should be published.

The authors have addressed most of the reviewer concerns, although in some cases in a rather perfunctory manner.

I remain concerned about the justification provided to chose the C4b2b nomenclature; indeed, this was "proposed" by four individuals well known in the field but never decided upon by the community. The term C4b2a is historically correct and is still predominant in the literature. It is a pity that this "proposal" has further complicated an already difficult nomenclature.

However, this is not the fault of these authors and at least now they note that this is not generally agreed (in fact should state that).

The defence of the stabilised convertases remains rather unconvincing (cross-linking and nanobody) but likely does not impact the data analysed.

In my opinion the Figure legends remain too brief and poorly informative - it would have been an easy fix to add a few lines to enable legend to properly explain figures.

Point-by-point reply

Reviewer #1:

1.1 First, the source of C4 or C4b used was only vaguely described (line 380). Were “serum-derived C4b and C2 originated from a single human subject or from mixed sera of multiple human subjects. This is particularly relevant for human C4 because of its extensive polymorphism of C4A and C4B isotypes with multiple allotypes in each protein. The polymorphic sites in C4 are overwhelmingly clustered at the TED domain and most human subjects consist of both acidic and basic isotypes. Unfortunately, cryoEM structures of the TED domain were not resolvable in this manuscript and thus their structural and functional relevance were obscure. In retrospect, the use of source proteins from single human subjects with minimal variations would have yielded more definitive or informative structures.

We purchased C4b from Complement Technology Inc., as indicated in line 390, which is indeed a mixture of C4A and C4B. A mixture of C4A and C4B obtained from pooled plasma is commonly used in structural studies (see [25, 32, 37, 38, 48]). Also, the set-up of this study was not focused on differentiating C4A and C4B. Moreover, differences in C4A and C4B localize to exposed areas in TED that do not form contact with other parts of the complex. Flexibility of the TED domain was anticipated, because variability in C3b-TED positioning was observed in the convertase structure of (C3bBb-SCIN)₂ (as shown by the very high B-factors). Most likely, the poor resolution in this area is foremost due to lack of interactions of TED, causing variations in its position, and to a lesser extent due to differences in C4Ab and C4Bb.

We added at line 393-396:

“Similar to Mortensen, *et al.* (2015 and 2016), we used C4b and C2 derived from pooled human serum (thus containing C4A and C4B isotypes) (Complement Technology, Inc.)...”

1.2 Second, dis-similar to many colleagues including the same senior author of this manuscript who published on complement structures, the heavy chain of C2 with serine proteinase and vWA domains (after activation was referred to as C2b instead of C2a. This creates unnecessary confusions, and feelings to many colleagues. Irrespective to the reasons of the nomenclature switch, it would be appropriate to add a footnote explaining the change of nomenclatures, or use a new set of names for the heavy and light chains of activated C2. The authors could acknowledge the limitations of using the reversed names for C2a and C2b in the Discussion

We follow the recommendations of Bohlson, Garred, Kemper and Tenner (2019). Originally, the large enzymatic fragment of C2 was called ‘C2a’ for ‘activated C2’; with the small fragment called ‘C2b’. For all other complement components, including the homologous factor B, ‘a’ refers to the small fragment and ‘b’ to the large fragment. Naming the large C2 fragment ‘C2b’ improves the consistency in nomenclature, in particular with respect to C2 and factor B. Unfortunately, the reference to Bohlson *et al.* was missing in the manuscript.

We added at line 68:

“(We follow the recommendations of Bohlson *et al.* [31], referring to the small C2 fragment as C2a and to the large C2 fragment as C2b.)”

We changed “... activation. These structures, ...” at line 301 into:

“... activation. Following the recommendations of Bohlson *et al.* (2019) [31], we refer to the large fragment of C2 as C2b, and not C2a (which originally indicated ‘activated’ fragment of C2). Hence, the fragments C2a and Ba vs. C2b and Bb of homologous proteases C2 and factor B refer to the equivalent parts. The structures of C4b2, C4b2b and C3-C4b2b, ...”

At line 129 we added after C2b:

“(previously reported as crystal structures of C2b and C2a, respectively)”

Reviewer #2:

The paper is well written and the figures are of good quality. Overall, I don't think the paper requires any real modification.

No modifications needed.

Reviewer #3 (Remarks to the Author):

3.1 One general concern is the decision to call the convertase C4b2b. The convertase was, from its first description in the 1960s, termed C4b2a. In recent years there has been a move from some quarters to "rationalise" by referring to the convertase as C4b2b; as the authors know, this is an area of continued disagreement in the field. This should be noted in the Introduction and the reason for deciding on C4b2b in the current Ms stated.

See reply 1.1

3.2 My major concern is around the generation of the complexes for analysis.

The C4b2 complex is generated by simply mixing commercially sourced plasma-derived C4b and C2 at 2.1 and 3.2 μm respectively in the presence of glutaraldehyde to non-specifically cross-link. How was this method arrived at and what is the evidence that the complex generated in this artificial way represents the labile proconvertase? Indeed, the gels shown in suppl Fig 1 suggest that there is very little C2 in the isolated final complex!

In fact, the method of complex formation of C4b2 is a, straightforward, solution equivalent of that what is happening on a targeted surface. Mortensen, *et al.* (2016) [32], generated C4b2 complexes in a similar way (though they stabilized the complex using Ni^{2+} (instead of Mg^{2+}) and did not use a crosslinking molecule).

Indeed, the efficiency of complex formation was low (in line with a low amount of C2 on the gel in suppl fig. 1). In the EM analysis we found particles containing C4b + C2 besides many particles consisting of C4b only. The generated C4b2 complex was in

overall organization consistent with the structural data obtained for the homologous C3bB.

We added at line 393-396:

“Similar to Mortensen, *et al.* (2015 and 2016)[25,32], C4b derived from pooled human serum (thus containing C4A and C4B isotypes) and C2 (Complement Technology, Inc.)...”

3.3 Conversion of proconvertase to convertase is mediated (I think - not clear in methods) by incubation of proconvertase with "preactivated C1s" but evidence of cleavage is not provided.

In contrast to pro-convertases, convertases are instable bimolecular complexes with a half-life time of 1-2 min [4,5]. C2 in the presence of activated C1s is cleaved completely at room temperature after 30 seconds. Thus, our method was focused on a very short (30 seconds) incubation time of C4 and C2 in the presence of activated C1s followed by flash-freezing the sample on the cryo-EM grid.

The resulting particles in cryo-EM unequivocally indicate cleavage of C4b2 forming C2b bound through its MIDAS in the VWA domain to the C-terminal end of C4b, as expected based on the structures of the alternative pathway proconvertase and SCIN-inhibited convertase.

We added Supplemental figure 6:

A gel showing complete cleavage of C2 into C2a and C2b by activated C1s within 30 seconds is added in panel A.

In panel B, the chromatogram shows cleavage of C4b2 (green) forming C4b2b and fragments C2a, C2b and C4b (black).

3.4 To generate convertase-substrate complexes, C3-C4b complexes are first generated by mixing the two proteins together with a cross-linking nanobody. Again, what is the evidence that this artificially generated complex adopts a conformation relevant to the natural convertase? This complex is then incubated with inactivated C2 and preactivated C1s to yield a nanobody-stabilised C4b2bC3 complex. once more, what is the evidence that this complex is representative of the native complex? supplementary figure 6 shows gels of the C3-C4b-nb complex but not of the complex incorporating C2b, essential for the characterisation.

To circumvent the problem of the short-life time of the convertase, we first pre-formed C4b-C3 complexes using the nanobody linker (thereby enhancing the binding affinity of C3) and then added C2 and preactivated C1s for 30 seconds and the immediately flash-froze the sample (as described in the material and methods). The correct formation of the convertase complex is evident from the C2b conformation and positioning onto C4b and binding of the scissile loop in a fully productive manner.

3.5 Minor points:

Figure legends are brief and would benefit from more explanation for non-experts; Some annoying abbreviations - for example, line 68 and elsewhere - "resp." for respectively, should be eliminated.

We checked the figure legends. We think that Figure legend 4 describe fully what is shown using descriptive phrasing (though indeed brief and without repeating the main text). We agree to add to Figure1-3 and 5 the following:

We added in the legends to figures 1-3 , line 725, 733 and 742
"180°-rotated "

We added in the legends to figure 1-3, line 728, 734 and 743
"from high (blue) to low (red), "

We added to the legend of figure 5, line 654 after account:
", showing that MG7-surface potentials of C4b-C3 are attractive and C4b-C3b are repulsive."

We have replaced "resp." by "respectively".

|